# Peritoneal Metastatic Gastric Cancer: Local Treatment Options and Recommendations

**Miklos Acs [1], Pompiliu Piso [2] and Gabriel Glockzin [3,***

[1]   Department of Surgery, University Medical Center Regensburg, 93053 Regensburg, Germany; miklos.acs@ukr.de
[2]   Department of Surgery, Krankenhaus Barmherzige Brueder Regensburg, 93049 Regensburg, Germany; pompiliu.piso@barmherzige-regensburg.de
[3]   Department of Surgery, Muenchen Klinik Bogenhausen, 81925 Munich, Germany
*   Correspondence: gabriel.glockzin@muenchen-klinik.de; Tel.: +49-89-9270-2011

**Abstract:** Peritoneal metastasis is a common finding in patients with advanced gastric cancer. Beyond systemic chemotherapy, additive local treatments such as cytoreductive surgery and intraperitoneal chemotherapy are considered an inherent part of different multimodal treatment concepts for selected patients with peritoneal metastatic gastric cancer. This review article discusses the role of cytoreductive surgery (CRS) and intraperitoneal chemotherapy, including HIPEC, NIPS, and PIPAC, as additive therapeutic options with curative and palliative intent.

**Keywords:** peritoneal metastasis; gastric cancer; cytoreductive surgery; intraperitoneal chemotherapy; HIPEC; NIPS; PIPAC

## 1. Introduction

Gastric cancer (GC) is still the fifth most common tumor entity and the fourth most common cause of cancer-related death worldwide. It occurs twice as often in men than in women [1]. Peritoneal metastasis is common in patients with advanced gastric cancer or disease progression and might be already present in 5–20% of patients at the time of surgery [2,3]. Moreover, peritoneal recurrence rates between 29% and 38% have been reported after curative resection of gastric cancer [4,5]. In a population-based data analysis of 5220 patients with gastric cancer from The Netherlands, the overall rate of peritoneal metastasis (PM) was 14% and the peritoneal metastasis only rate was 9%. The median survival of patients with peritoneal metastatic gastric cancer (pmGC) in this national registry was 4 months [6]. The standard of care for patients with metastatic gastric cancer is systemic chemotherapy and immunotherapy, or, in some cases, best supportive care [7–9].

The French and ESMO (European Society for Medical Oncology) guidelines recommend palliative intravenous chemotherapy only in patients with pmGC [8,10]. The 2023 National Comprehensive Cancer Network (NCCN) guidelines recommend that all patients with metastatic gastric cancer, including peritoneal metastasis, should be treated with chemoradiation, systemic therapy, or best supportive care. Accordingly, gastric resection should only be performed with palliative intent in the case of symptomatic disease refractory to conservative treatment [11].

Whether the addition of gastrectomy to chemotherapy improves survival for patients with advanced gastric cancer was investigated in the REGATTA phase III randomized controlled trial. In this trial, gastrectomy followed by chemotherapy did not show any survival benefit compared with chemotherapy alone [12].

Since the main cause of peritoneal metastasis in GC is the dissemination of free tumor cells in the peritoneal cavity, additional regional therapy modalities that aim to target these tumor cells could provide oncological benefit [13]. Nevertheless, due to the poor prognosis of patients with pmGC and inconsistent published data, the additional use of

locoregional treatment options as an inherent part of an interdisciplinary multimodality treatment concept remains under debate.

This review aimed to discuss the current evidence, ongoing trials, and recommendations for local treatment options, such as hyperthermic intraperitoneal chemotherapy (HIPEC), neoadjuvant intraperitoneal and systemic chemotherapy (NIPS), and pressurized intraperitoneal aerosol chemotherapy (PIPAC), in peritoneal metastatic gastric cancer patients.

## 2. Cytoreductive Surgery and Hyperthermic Intraperitoneal Chemotherapy (HIPEC)

The concepts of cytoreductive surgery (CRS) and hyperthermic intraperitoneal chemotherapy (HIPEC) have existed since the late 1980s, with the aim of surgical resection of all intra-abdominal macroscopic disease. The completeness of the cytoreduction (CC) score defines the level of success of the surgery [14]. Additionally, HIPEC provides the cytotoxic effect of hyperthermia, as well as regional dose intensification, through chemotherapy perfusion in the abdominal cavity, to consolidate the effect of complete macroscopic cytoreduction and to treat free abdominal tumor cells and any potentially remaining microscopic disease [15–17]. The intrabdominal tumor burden can be easily calculated using the Peritoneal Cancer Index (PCI), which is given as a numerical score ranging from 0 to 39 [18].

In a recently published meta-analysis including 691 patients with peritoneal metastatic gastric cancer from two RCTs and eight NRCTs, Zhang et al. demonstrated an overall survival benefit of 4.67 months for patients that underwent CRS and HIPEC. Nevertheless, there was no statistically significant difference in the 3-year overall survival rates in the two RCTs. Complete macroscopic cytoreduction (CC-0/1) was crucial to the patient's outcome. CRS and HIPEC did not increase the complication rates [19]. Chen et al. compared survival rates of 1367 patients treated with CRS plus HIPEC versus CRS alone in eight RCTs and two NRCTs. There was a significant survival advantage after 3 years in patients that underwent CRS and HIPEC (RR = 0.63, 95% CI 0.36–1.07, $p < 0.05$). However, there was no significant difference in the 1-year and 5-year overall survival [20]. Consistent with these results, Desiderio et al. reported no survival advantage in patients with gastric cancer and proven peritoneal metastasis treated with neoadjuvant systemic chemotherapy and gastrectomy plus HIPEC in comparison to patients that underwent neoadjuvant chemotherapy and gastrectomy without HIPEC. Nevertheless, the HIPEC group showed a prolonged median survival of 4 months [21].

Bonnot et al. reported improved survival rates for patients with pmGC treated with CRS plus HIPEC versus CRS alone. Data from 277 patients from a prospective database were included and propensity score analysis was performed. Patient characteristics did not differ in the two groups, except for the median Peritoneal Cancer Index (PCI), which was higher in the CRS/HIPEC group (6 vs. 2). A median survival of 18.8 versus 12.1 months ($p = 0.002$) was found after IPTW adjustment of the groups. The 5-year OS rates were 19.9 and 6.4%, respectively. There was no statistically significant difference in the morbidity rates (CTCAE > 3) and 90-day mortality between the two groups [22]. Data from 88 patients with pmGC treated with CRS plus HIPEC, published by the Spanish Group of Peritoneal Oncologic Surgery (GECOP), showed a median survival of 21.2 months and a 3-year overall survival rate of 30.9%. Major complications (Dindo–Clavien °III/IV) occurred in 31% of the patients. The mortality rate was 3.4%. For the subgroup of patients with PCI < 6, the median survival time was 26.1 months and the 5-year overall survival rate reached 46.8%. The multivariate analysis identified PCI > 6 to be the only negative predictive factor for survival [23]. This observation was supported by the published data of 235 patients with pmGC from the German HIPEC registry. The median survival was 18 months for patients with a PCI between 0 and 6, 12 months for patients with a PCI from 7 to 15, and only 5 months for patients with a PCI higher than 15 [24]. The Italian Peritoneal Malignancies Oncoteam reported a median overall survival of 20.2 months after CRS and HIPEC in 91 patients with pmGC from a retrospective registry. Patients with PCI ≤ 6 showed

a significantly better median survival than patients with a higher PCI (44.3 months vs. 13.4 months) [25].

Published data have also shown that long-time survival and cure can be achieved in selected patients with pmGC after CRS and HIPEC. Brandl et al. analyzed 28 patients from an international PSOGI cohort who survived for more than five years. The median survival in this subgroup of patients was 11 years. In 12 of 28 patients, disease recurrence occurred after a median time of 9.6 years. The overall survival time in these 12 patients was still 8.8 years. Positive predictive factors for long-term survival were a PCI $\leq$ 6 and complete macroscopic cytoreduction (CC-0) [26].

Badgwell et al. reported a median OS of 16.1 months and a 3-year OS of 28% in 20 patients with at least positive peritoneal cytology or pmGC with a median PCI of 2 (0–13) enrolled in a prospective phase II clinical trial. All patients received systemic chemotherapy and laparoscopic HIPEC before cytoreductive surgery including gastrectomy, D2 lymphadenectomy, and HIPEC [27].

The aim of the first published prospective randomized controlled phase III trial was to evaluate the efficacy and safety of CRS plus HIPEC versus CRS alone in 68 patients with pmGC. PCI did not impact patient inclusion, and therefore ranged from 2 to 36. The median PCI was PCI 15 and the rate of complete macroscopic cytoreduction (CC-0/1) was 58.8% in both groups. Patients treated with CRS plus HIPEC recorded a median survival of 11 months versus 6.5 months in the CRS only group. Synchronous peritoneal metastasis, complete macroscopic cytoreduction, and at least six cycles of systemic chemotherapy were found to be independent factors for improved survival in the multivariate analysis [28]. The GYMSSA trial, comparing CRS and HIPEC followed by systemic chemotherapy to chemotherapy alone in patients with pmGC, was prematurely terminated after the inclusion of 16 patients. In seven of the nine patients in the CRS/HIPEC group, complete macroscopic cytoreduction was achieved. Median survival was 11.3 months in the treatment group and 4.3 months in the chemotherapy only group [29]. The German prospective randomized phase III GASTRIPEC trial investigated perioperative systemic chemotherapy plus CRS alone versus perioperative systemic chemotherapy plus CRS and HIPEC in patients with pmGC. The trial was prematurely stopped after the recruitment of 105 of the 180 calculated patients. A total of 52 of these patients were randomized for CRS only and 53 for CRS plus HIPEC. Within the sample, 55 of the 105 patients did not undergo surgery due to disease progression or death. Thus, the results of data analysis are limited. Nevertheless, there was no statistically significant difference in OS between the two groups (14.9 months in both groups), but a statistically significant trend for improved median progression-free survival (7.1 vs. 3.5 months) and median metastasis-free survival (10.2 vs. 9.2 months) was found for the CRS plus HIPEC group. The perioperative 30-day morbidity (Dindo–Clavien °III/IV) was similar in both groups [30].

The actual German guidelines for gastric cancer published in 2019 do not recommend treatment of patients with pmGC with CRS and HIPEC outside of clinical trials [7]. The ESMO guidelines also recommend the inclusion of patients in clinical trials for CRS and HIPEC [8]. The UK National Institute for Health and Care Excellence stated in 2021 that there is data supporting CRS and HIPEC for selected patients with peritoneal metastasis, but the evidence for its specific benefit over systemic chemotherapy is absent [31]. The NCCN guidelines published in 2022 indicated that the role of surgery in patients with gastric cancer and positive cytology is uncertain. Surgery might be performed after systemic chemotherapy. CRS plus HIPEC in patients with pmGC was not addressed [9]. In the online published version 1.2023, it was stated that the use of HIPEC should only be considered for selected patients in the framework of ongoing clinical trials [11]. The Chicago Consensus Working Group recommends cytoreductive surgery with or without HIPEC in selected patients with limited peritoneal metastasis of gastric origin (PCI < 6) as an integrated part of a multimodal treatment concept with curative intent. Complete macroscopic cytoreduction should be achievable. Due to the results of the REGATTA trial, palliative gastrectomy is not recommended [32]. Actual guidelines or recommendations from the Peritoneal Surface

Oncology Group International (PSOGI) regarding CRS with or without HIPEC in patients with pmGC are not available.

There are fourteen ongoing clinical trials registered at ClinicalTrials.com investigating HIPEC treatment in patients with peritoneal metastatic and/or locally advanced gastric cancer with therapeutic or prophylactic intent [33]. The ongoing clinical trials for patients with pmGC are summarized in Table 1. Clinical trials investigating prophylactic HIPEC in patients with a high risk of development of peritoneal metastasis, such as PREVENT, GOETH, and GASTRICHIP, are not included [34–36]. The ongoing prospective randomized phase III PREVENT trial is investigating perioperative chemotherapy with FLOT plus surgery with or without cisplatin-based HIPEC in patients with primary resectable gastric cancer. The proposed number of patients to be included is 200 [34]. However, the only European ongoing RCT for patients with proven pmGC is the PERISCOPE II trial (NCT03348150) [37].

**Table 1.** Ongoing clinical trials for the treatment of patients with pmGC.

| Trial ID | Phase | Allocation | N | Start | End (est.) | Land | Status |
|---|---|---|---|---|---|---|---|
| NCT02381847 | phase III | randomized | 60 | 2015 | 2020 | China | unknown |
| NCT03179579 | phase III | randomized | 88 | 2017 | 2022 | China | unknown |
| NCT03023436 | phase III | single-arm | 220 | 2016 | 2022 | China | unknown |
| NCT03348150 | phase III | randomized | 182 | 2017 | 2029 | The Netherlands | recruiting |

### 3. Neoadjuvant Treatment: Intraperitoneal Systemic Chemotherapy (NIPS)

The INPACT trial, a prospective randomized multicenter phase II clinical trial, investigated repeated normothermic intraperitoneal chemotherapy as an additive treatment after gastrectomy in patients with gastric cancer and peritoneal metastasis or positive cytology, and proved its safety and efficacy. Nevertheless, there was no significant advantage of intraperitoneal over systemic chemotherapy in overall survival [38]. The prospective randomized PHOENIX-GC trial compared 114 patients with pmGC treated with normothermic intraperitoneal chemotherapy plus systemic chemotherapy to 50 patients treated with systemic chemotherapy only. The median overall survival was 17.7 months and 15.2 months, respectively ($p = 0.08$). The 3-year OS rate was 21.9% vs. 6.0%, suggesting a survival advantage for the IP treatment group [39]. The feasibility and safety of combined intraperitoneal and systemic chemotherapy has also been confirmed by a prospective phase II trial. The median OS survival of the 35 included patients was 17.6 months [40]. Another recently published phase II trial reported a median survival of 19.4 months and a 1-year OS rate of 73.6% after combined systemic and intraperitoneal chemotherapy in patients with pmGC. The grade III/IV hematologic and non-hematologic morbidity rates were 43% and 47%, respectively [41].

The idea of NIPS is to use repeated normothermic intraperitoneal chemotherapy plus systemic chemotherapy as a neoadjuvant treatment, followed by gastrectomy and D2 lymphadenectomy. Shinkai et al. reported a regression rate of 73.3% after NIPS in 17 patients included in a single-arm prospective phase II trial. The median OS was 23.9 months, and the 1-year survival rate was 82.4% [42]. Comparable survival data were published by Saito et al. after the treatment of 44 patients with pmGC or positive cytology with combined systemic chemotherapy with S1 and oxaliplatin and intraperitoneal chemotherapy with paclitaxel. The 1-year overall survival rate was 79.5%. The median overall survival was 25.8 months. Twenty patients (45%) became resectable due to regression of peritoneal metastasis [43].

The Peritoneal Surface Oncology International (PSOGI) proposed a combined treatment concept including NIPS followed by complete macroscopic cytoreduction and HIPEC in patients with pmGC or positive cytology [44]. Yonemura et al. reported a 5-year OS

of 9.6% and a 10-year OS of 5.0% in 419 patients with pmGC treated with NIPS followed by cytoreductive surgery. A total of 255 patients received intraoperative HIPEC. After complete macroscopic cytoreduction (CC-0), the 5-year and 10-year survival rates reached 14.5% and 8.3%. After incomplete cytoreduction (CC-1), the 5-year and 10-year survival rates were 1.8% and 0%, respectively. All patients with incomplete cytoreduction died within 6 years. The median survival was 20.5 versus 12 months [45]. A prospective phase II clinical trial that enrolled 67 patients with pmGC proved the safety and efficacy of NIPS. A 1-year overall survival rate of 67.2% and a median overall survival of 19.3 months was reported. A total of 42 patients underwent conversion surgery [46]. A meta-analysis of eight retrospective studies evaluated the efficacy of NIPS followed by surgery in comparison to NIPS only in 373 patients with pmGC. There was a statistically significant survival advantage for patients that underwent surgery after NIPS [47].

The ongoing prospective randomized phase III multicenter DRAGON II trial is investigating the effect of combined neoadjuvant laparoscopic HIPEC and systemic chemotherapy followed by gastrectomy with D2 lymphadenectomy plus HIPEC and five cycles of adjuvant chemotherapy versus surgery without HIPEC followed by eight cycles of adjuvant systemic chemotherapy in 326 patients with pmGC. The study endpoints are 5-year PFS, 5-year OS, peritoneal metastasis rate, and perioperative morbidity [48]. Another prospective randomized controlled phase III trial was conducted in China to compare NIPS plus neoadjuvant systemic chemotherapy to neoadjuvant chemotherapy only for the treatment of pmGC in 238 patients [49]. The preliminary or final results of these two trials have not yet been published.

## 4. Pressurized Intraperitoneal Aerosol Chemotherapy (PIPAC)

Intraperitoneal delivery of cytostatic drugs as pressurized intraperitoneal aerosol chemotherapy (PIPAC) has emerged as a feasible therapeutic option in patients with peritoneal malignancies of primary or secondary origin [50–52]. This approach combines the advantages of the homogenous distribution of an aerosol and low-dose intraperitoneal chemotherapy, such as a high intraperitoneal concentration with a low systematic concentration. When first introduced by a German group in 2012, PIPAC was used for the treatment of tumor patients with end-stage disease [53]. Over time, PIPAC was used to treat gastric cancer patients with neoadjuvant intent or as a part of an adjuvant treatment. Unfortunately, to date, there have been no phase III studies or randomized controlled trials considering the scope of PIPAC. Most of the studies on this treatment have been prospective or retrospective cohort investigations.

In a recent meta-analysis conducted by Di Giorgio et al., a pathological response of 69% (95% CI 0.60–0.77) after PIPAC was reported, ranging between 53.8 and 75% in pmGC patients. The study included 1990 patients who underwent 4719 PIPAC procedures, in which the overall proportion of patients who completed three or more cycles of PIPAC with gastric cancer was 34%. In the same study, an estimated 12-month survival from first PIPAC for gastric cancer was 25%, with a median overall survival ranging between 4 and 19.1 months [53]. Nadiradze et al. reported an overall survival of 15.4 months after PIPAC therapy in 24 patients with therapy-resistant pmGC. In 50% of patients, a histological tumor response was able to be obtained [54]. Another retrospective analysis showed a median survival of 19.1 months after at least three cycles of PIPAC in 42 patients with irresectable pmGC. Six patients became resectable after PIPAC treatment [55]. Casella et al. retrospectively analyzed the outcome of 74 patients with synchronous peritoneal metastasis of gastric origin treated with PIPAC and systemic chemotherapy. The mean PCI was 16 (range 8–26). The median overall survival from the time of diagnosis was 19.6 months and from the first PIPAC was 10.5 months. Eleven of 74 patients could be operated on due to response to the combined treatment [56]. Sindayigaya et al. reported a median overall survival of 11 months in 144 patients with pmGC scheduled for PIPAC and 16 months for the subgroup of patients that received at least three cycles of PIPAC. The ten patients (7%)

that underwent CRS and HIPEC after PIPAC therapy recorded a median overall survival of 15 months [57].

In a Russian prospective phase II trial that included 31 patients with pmGC, the patients received four cycles of systemic chemotherapy followed by PIPAC every 6 weeks, with two cycles of systemic chemotherapy between the PIPAC cycles. The overall survival was 13 months and 60% of the patients showed a major pathologic response [58]. Struller et al. were able to show stable disease or disease regression in 25 patients with therapy-resistant pmGC treated with PIPAC every six weeks for three cycles within a prospective phase II trial. The median overall survival was 6.7 months [59]. The prospective phase II PIPAC-OPC2 trial investigated PIPAC treatment in patients with peritoneal metastasis from different origins. The median survival of the subgroup of pmGC patients was 7.4 months [60]. Kryh-Jensen et al. retrospectively analyzed data from the prospective PIPAC-OPC1 and PIPAC-OPC2 studies, considering long-term survival after PIPAC for the treatment of peritoneal metastasis of different origins. Nine of 39 patients with pmGC (23%) survived for at least 21 months, defined as long-term survival for this group of patients. The median overall survival after first PIPAC was 7.8 months [61]. Ellebaek et al. reported a median survival of 11.5 months from the time of diagnosis and 4.7 months after the first PIPAC procedure in a smaller cohort of twenty pmGC patients from the same database. Ten of the twenty patients completed the scheduled three cycles of PIPAC [62].

The outcome data of the above-mentioned studies is summarized in Table 2.

**Table 2.** Outcome after PIPAC therapy in patients with pmGC. * PM of different origin, ** pmGC only, n.a.: not assessed.

| Author | Study Design | Morbidity, SAE [%] | Mortality [%] | Histological Response [%] | Median OS [Months] |
|---|---|---|---|---|---|
| Di Giorgio et al. [53] | meta-analysis | 4 | 1.3 | 69.2 | 12.3 |
| Nadiradze et al. [54] | retrospective | 12.5 | 8.3 | 50 | 15.4 |
| Alyami et al. [55] | retrospective | 6.1 | 4.7 | n.a. | 19.1 |
| Casella et al. [56] | retrospective | 4 | 0 | n.a. | 10.5 |
| Sindayigaya et al. [57] | retrospective | 4.9 | 1.4 | 73 | 11 |
| Khomyakov et al. [58] | phase II | 3.2 | 0 | 60 | 13 |
| Struller et al. [59] | phase II | 12 | 0 | 75 | 6.7 |
| Graversen et al. [60] | phase II | 4.5 | 4 | 61 * | 7.4 ** |

Recently, an international PIPAC consensus meeting was held by dedicated experts to limit the heterogeneity of treatment protocols. Accordingly, 90.9% of the experts agreed to use the combination of doxorubicin ($2.1$ mg/m$^2$) and cisplatin ($10.5$ mg/m$^2$) for PIPAC therapy, while 72.7% supported oxaliplatin at 120 mg/m$^2$. A 25% dose reduction can be considered in frail patients (90 mg/m$^2$). Mitomycin-C and Nab-paclitaxel were proposed as other alternative drugs for PIPAC [63].

The PIPAC VER-One trial is an ongoing prospective randomized phase III clinical trial that compares systemic chemotherapy plus PIPAC to systemic chemotherapy only as a treatment for patients with gastric cancer and positive cytology or limited peritoneal metastasis (PCI $\leq$ 6). The primary endpoint is the secondary resectability rate. Moreover, overall survival, progression-free survival, disease-free survival, morbidity according to CTCAE, histological response, quality of life, and cost effectiveness are being evaluated [64]. The prospective phase II PIPAC EstoK 01 trial investigating PIPAC plus intravenous chemotherapy versus systemic chemotherapy only in patients with pmGC with a Peritoneal Cancer Index (PCI) > 8 had to be stopped due to an elevated number of bowel obstructions, after having included 64 patients. The initial proposed number of patients to be included was 94 [65].

## 5. Laparoscopic HIPEC with Palliative Intent

Peritoneal metastasis is often associated with malignant ascites that, in some cases, cannot be managed using conservative treatment such as systemic chemotherapy, immunotherapy, medical treatment, and paracentesis. Symptomatic ascites seriously impair the quality of life of tumor patients [66]. Interestingly, regression of malignant ascites has been described as a positive side-effect of HIPEC treatment [67]. Valle et al. published the data of 12 patients with peritoneal metastasis from different origins and refractory malignant ascites treated with palliative laparoscopic HIPEC without cytoreductive surgery (CRS). The malignant ascites completely disappeared in ten out of twelve patients. Two patients developed recurrent ascites 124 and 283 days after palliative laparoscopic HIPEC. The grade III/IV morbidity and the mortality rate were both 0%. Nevertheless, there were only two patients included with pmGC and their postoperative survival was short, at only 16 and 50 days, respectively [68]. Facchiano et al. reported a regression of ascites in five patients with unresectable pmGC treated with laparoscopic HIPEC [69]. Another retrospective analysis comparing 45 patients with >3000 mL of malignant ascites of different origins treated with repeated laparoscopic HIPEC followed by systemic chemotherapy to 35 patients treated with systemic chemotherapy only showed a short-term total effective rate of 91% vs. 40% in the control group. Total response has been defined as the disappearance of ascites in >4 weeks and partial response as a reduction of ascites >50% for at least 4 weeks. There was no statistically significant difference regarding in the treatment-associated adverse events between the groups [70].

Zao et al. reported a complete or partial ascites remission rate of 70.6% in 26 patients treated with two cycles of whole-body hyperthermia and HIPEC versus 33.3% in 21 patients of the control group treated with systemic chemotherapy [71]. Comparable data was published by Yarema et al., showing ascites recurrence in only two of ten patients treated with palliative surgery and HIPEC [72]. In a prospective randomized clinical trial published by Zhang et al. investigating palliative laparoscopic surgery followed by HIPEC and systemic chemotherapy with (treatment group, $n = 60$) and without additional immunotherapy (control group, $n = 60$), the objective remission rates of the ascites were 58% and 35% ($p = 0.01$), respectively [73].

## 6. Discussion

It is certain that systemic chemotherapy or immunotherapy are the standard of care in patients with peritoneal metastasis of gastric origin. Nevertheless, additional local treatments, including surgery and intraperitoneal chemotherapy, might improve survival in selected patients. Although the treatment of patients with pmGC with cytoreductive surgery and hyperthermic intraperitoneal chemotherapy within clinical trials is recommended by most guidelines, clinical trials are often not available (Table 1). Thus, we propose the following treatment algorithm for selected patients with gastric cancer and limited peritoneal metastasis (Figure 1). All patients should be discussed in a multidisciplinary tumor board prior to initial therapy, and the chosen multimodal treatment concept must be evaluated regarding the expected results and adapted if necessary. Moreover, patient data should be prospectively recorded and regularly analyzed with regard to feasibility, complications, and patient outcome.

There is good evidence from prospective randomized clinical trials for the efficacy of neoadjuvant systemic chemotherapy with FLOT in patients with limited metastatic and locally advanced gastric cancer. Moreover, this regimen showed superiority over other drug combinations in this clinical setup [74–76]. Based on this data, FLOT is recommended as the standard of care for neoadjuvant treatment of gastric cancer patients by national and international guidelines [7,8]. Thus, the FLOT regimen might also be preferred for systemic treatment of patients with pmGC with neoadjuvant intent. However, tumor histology and molecular tumor characteristics play a pivotal role in the prognosis and response to systemic therapy. Tumor characteristics such as positive HER2 status, MSI, or expression of Claudin18.2 already determine additional therapeutic options in patients with gastric can-

cer [77]. Based on individual patient and tumor characteristics, alternative chemotherapy regimens, targeted therapy, and immunotherapy might be considered for drug selection in order to reach the maximal response and tumor regression in a neoadjuvant setting [77,78].

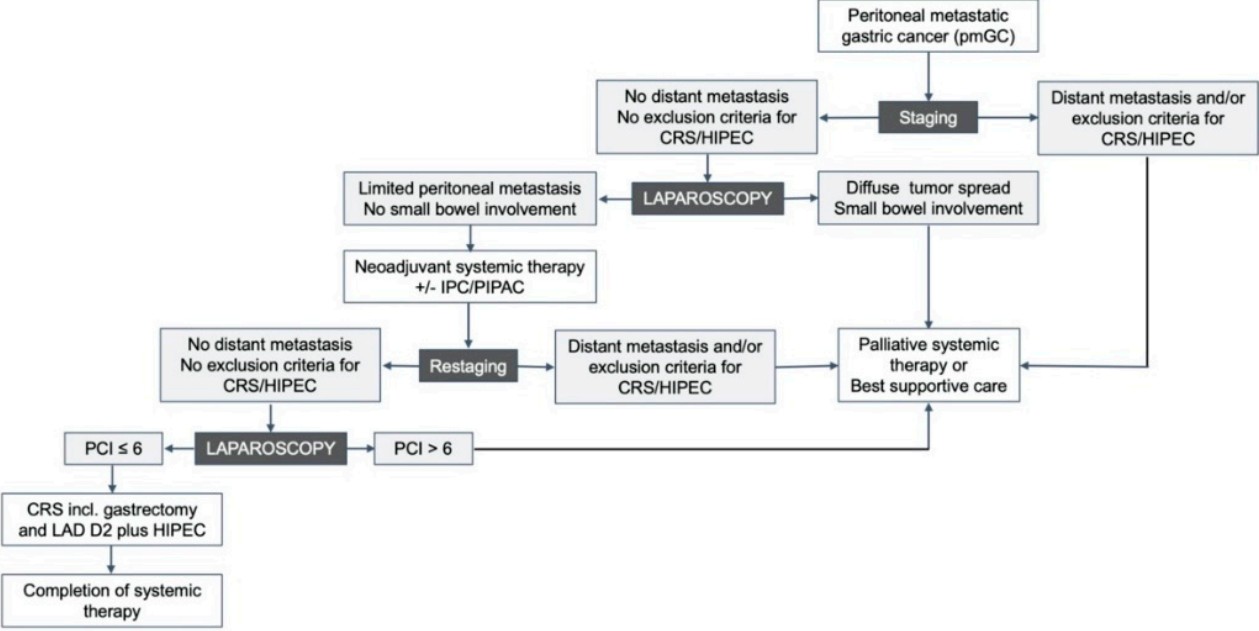

**Figure 1.** Proposed treatment algorithm for patients with peritoneal metastatic gastric cancer (pmGC).

Published studies have shown consistently that the peritoneal distribution of tumor deposits and the complete removal of all visible intraperitoneal tumor nodules during cytoreductive surgery play a pivotal role in the oncological outcome of treated patients. Considering the available survival data from prospective clinical trials and retrospective analyses, the cut-off value for patients with pmGC that qualify for CRS and HIPEC seems to be a Peritoneal Cancer Index (PCI) of 6. Moreover, complete macroscopic cytoreduction (CC-0) has been shown to be an independent positive predictive marker for improved survival in almost all published multivariate analyses.

In contrast, the optimal HIPEC regimen for patients with pmGC is still a matter of debate, and published data do not support clear recommendations. Numerous different drugs, dosages, and perfusion protocols have been reported as showing comparable efficacy and safety profiles. Selected representative published HIPEC regimens from four randomized controlled trials, prospective studies, and retrospective analyses are summarized in Table 3.

Considering national guidelines and local standard operating procedures, additional local treatment with NIPS or PIPAC might be discussed as an integrated part of a neoadjuvant intended interdisciplinary treatment concept. Alyami et al. showed that repeated PIPAC may lead to tumor regression and allow for CRS plus HIPEC in selected patients with peritoneal metastasis. Twenty-six out of 146 patients given palliative PIPAC treatment were scheduled for CRS and HIPEC. In 21 patients (14.4%), complete macroscopic cytoreduction was achieved [55]. However, the published data do not allow for clear recommendations for PIPAC in the neoadjuvant treatment of pmGC patients. Results from prospective RCTs providing high level evidence are not available.

**Table 3.** Selected representative published HIPEC regimens for pmGC.

| Author | HIPEC Regimen | Duration of Perfusion | RCT |
|---|---|---|---|
| Badgwell et al. [79] | mitomycin C 30 mg + cisplatin 200 mg intravenous sodium thiosulfate | 60 min | no |
| Glehen et al. [80] | mitomycin C 30–50 mg/m$^2$ ± cisplatin 50–100 mg/m$^2$ | 60–120 min | no |
| Glehen et al. [80] | oxaliplatin 360–460 mg/m$^2$ ± irinotecan 100–200 mg/m$^2$ ± intravenous 5-FU/FA | 30 min | no |
| Koemans et al. [37] | oxaliplatin 460 mg/m$^2$ + docetaxel 50 mg/m$^2$ | 30 + 90 min | yes |
| Piso et al. [81] | cisplatin 75 mg/m$^2$ + doxorubicin 15 mg/m$^2$ | 60 min | no |
| Rau et al. [30] | mitomycin C 15 mg/m$^2$ + cisplatin 75 mg/m$^2$ | 60 min | yes |
| Rudloff et al. [29] | oxaliplatin 460 mg/m$^2$ + intravenous 5-FU/FA | 30 min | yes |
| Shen et al. [82] | mitomycin C 30 mg + additional 10 mg at 60 min | 120 min | no |
| Van der Kaaij [83] | oxaliplatin 460 mg/m$^2$ + docetaxel 0, 50, 75 mg/m$^2$ | 30 + 90 min | no |
| Yang et al. [28] | mitomycin C 30 mg + cisplatin 120 mg | 60–90 min | yes |
| Yarema et al. [72] | mitomycin C 12.5 mg/m$^2$ + cisplatin 75 mg/m$^2$ + intravenous 5-FU | 90 min | no |
| Yarema et al. [84] | mitomycin C 10–15 mg/m$^2$ ± cisplatin 75 mg/m$^2$ ± intravenous 5-FU<br>cisplatin 75 mg/m$^2$ + doxorubicin 15 mg/m$^2$<br>oxaliplatin 460 mg/m$^2$ + intravenous 5-FU | 30–90 min | no |
| Yonemura et al. [85] | mitomycin C 30 mg + cisplatin 300 mg + etoposide 150 mg | 60 min | no |
| Yonemura et al. [45] | mitomycin C 12.5 mg/m$^2$ + cisplatin 50 mg/m$^2$ | 60 min | no |

A novel treatment strategy with laparoscopic hyperthermic intraperitoneal chemotherapy for patients with pmGC was developed at the University of Texas, at the MD Anderson Cancer Center. Inclusion of patients was based upon strict eligibility criteria, such as limited peritoneal metastasis and/or positive peritoneal cytology, ECOG performance status ≤2, and adequate renal, hematologic, and liver function. Patients with metastases other than peritoneal were excluded from the treatment protocol, which consisted of first-line chemotherapy, re-staging, and laparoscopic HIPEC in the case of regression or stable disease. After laparoscopic HIPEC, the patients were re-staged again to decide on further treatment options: repeated laparoscopic HIPEC, CRS/HIPEC including gastrectomy, second-line systemic chemotherapy, or inclusion in a phase I clinical trial [86]. The analysis included 44 patients who underwent 71 laparoscopic HIPEC procedures, and illustrated the feasibility of neoadjuvant laparoscopic HIPEC [79,86]. These patients underwent curative-intent multimodal therapy as part of a standardized algorithm for patients with pmGC.

Yonemura et al. reported a statistically significant reduction of PCI after one cycle of neoadjuvant intended laparoscopic HIPEC and after two cycles of laparoscopic HIPEC plus three cycles of NIPS. After this combined treatment, complete cytoreduction was achieved in 30 of 52 patients (56.7%) [87].

Current data from several ongoing RCTs legitimize the use of CRS and HIPEC to improve historically poor outcomes in patients with pmGC. However, many questions still need to be clarified, including the optimal timing, treatment regimen, dosing, and treatment algorithm. The eagerly awaited PERISCOPE II multicenter randomized controlled trial will definitively determine whether gastric cancer patients with limited peritoneal dissemination and/or tumor positive peritoneal cytology treated with systemic chemotherapy, gastrectomy, CRS, and HIPEC show a survival benefit over patients treated with palliative systemic chemotherapy only [37].

In patients with therapy refractory symptomatic malignant ascites due to peritoneal metastasis of gastric origin, additional local treatment options, such as laparoscopic HIPEC or PIPAC, should be considered to improve quality of life. However, patient-related factors such as comorbidities and prognosis, the pre-interventional quality of life, and the individual risk of perioperative and postoperative complications should be considered in decision making.

## 7. Conclusions

There is evidence that the local treatment of peritoneal metastasis as part of an interdisciplinary treatment concept may improve survival and quality of life in selected patients with peritoneal metastatic gastric cancer. Thus, all patients should be discussed in an interdisciplinary tumor board at the time of diagnosis. Consistent patient selection plays a crucial role in the efficacy of the different local treatment options and helps to avoid unnecessary adverse events.

Further clinical trials are needed to investigate the therapeutic potential of additive local treatment for peritoneal metastasis of gastric origin in the survival and improvement of quality of life of patients.

**Author Contributions:** Conceptualization: P.P. and G.G.; writing: M.A. and G.G.; review and editing: P.P. and G.G. All authors have read and agreed to the published version of the manuscript.

**Funding:** This research received no external funding.

**Conflicts of Interest:** The authors declare no conflicts of interest.

## Abbreviations

| | |
|---|---|
| CC | completeness of cytoreduction score |
| CRS | cytoreductive surgery |
| CTCAE | Common Terminology Criteria for Adverse Events |
| ESMO | European Society for Medical Oncology |
| FA | folinic acid |
| FLOT | 5-FU/leucovorin/oxaliplatin/docetaxel |
| 5-FU | 5- fluorouracil |
| HER2 | human epidermal growth factor receptor 2 |
| HIPEC | hyperthermic intraperitoneal chemotherapy |
| IPC | intraperitoneal chemotherapy |
| IPTW | inverse probability of treatment weighting |
| LAD | lymphadenectomy |
| MFS | metastasis-free survival |
| MMC | mitomycin C |
| MSI | microsatellite instability |
| NCCN | National Comprehensive Cancer Network |
| NICE | National Institute for Health and Care Excellence |
| NIPS | neoadjuvant intraperitoneal systemic chemotherapy |
| RCT | randomized controlled trial |
| NRCT | non-randomized controlled trial |
| OS | overall survival |
| PCI | Peritoneal Cancer Index |
| PFS | progression-free survival |
| PIPAC | pressurized intraperitoneal aerosol chemotherapy |
| pmGC | peritoneal metastatic gastric cancer |
| PR | partial response |
| RCT | randomized controlled trial |
| TR | total response |

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
