# Peer review of "Peritoneal Metastatic Gastric Cancer: Local Treatment Options and Recommendations"

_curroncol, doi:10.3390/curroncol31030109_

Round 1

Reviewer 1 Report

Comments and Suggestions for Authors

In this paper, the researchers address the important problem of advanced gastric cancer, which unfortunately is common ground between radiologists, for diagnostics, surgeons for the technical-operational part, and oncologists who deal with the disease with chemotherapy with neo-treatment. adjuvant before surgery and subsequently adjuvant post-operatively. In the introduction I believe it would be appropriate to say a few words on the histology of the tumors that need to be treated, given that we know, for example, that linitis is the neoplasm that responds worst to any type of therapy. Furthermore, at the time of diagnosis it is possible to immediately carry out microbiology to start patients with advanced neoplasia on neoadjuvant FLOT or DOC (doi.org/10.1016/j.suronc.2019.10.002). In the method they conduct a review of many articles published on the topic, taking into consideration essential endpoints, such as FLOT; HIPEC, PIPAC and PCI. They do not neglect to consider immunotherapy which has appeared on the horizon quite recently and HER 2 which provides a specific treatment. All these are already known topics but which are willingly reread in a general examination. In our hospital we have conducted a study with good results in advanced gastric tumors (doi.org/10.1097/CAD.0000000000000877 to be cited in the bibliography) which perhaps adds something to the types of therapy proposed. We absolutely agree that to treat these patients it is necessary to organize multidisciplinary meetings to agree on the best diagnostic/therapeutic approach through brain storming; and during the established path, the discussion must be updated once the results are achieved. The paper must be reviewed in light of these lines to be accepted. The bibliography is good, the tables are explanatory

Comments on the Quality of English Language

there are few errors to review

Author Response

Response to reviewers’ comments

Reviewer 1:

“In this paper, the researchers address the important problem of advanced gastric cancer, which unfortunately is common ground between radiologists, for diagnostics, surgeons for the technical-operational part, and oncologists who deal with the disease with chemotherapy with neo-treatment. adjuvant before surgery and subsequently adjuvant post-operatively.”

  1. “In the introduction I believe it would be appropriate to say a few words on the histology of the tumors that need to be treated, given that we know, for example, that linitis is the neoplasm that responds worst to any type of therapy. Furthermore, at the time of diagnosis it is possible to immediately carry out microbiology to start patients with advanced neoplasia on neoadjuvant FLOT or DOC (doi.org/10.1016/j.suronc.2019.10.002).”

We agree with the reviewer that histology and molecular biology of gastric cancer plays a pivotal role for patient outcome. Due to the main topic of our manuscript, we have not addressed this issue in the introduction. Nevertheless, this issue is discussed in conjunction with the proposed treatment algorithm. Due to the reviewers comments we revised this section of discussion.

  1. “In the method they conduct a review of many articles published on the topic, taking into consideration essential endpoints, such as FLOT; HIPEC, PIPAC and PCI. They do not neglect to consider immunotherapy which has appeared on the horizon quite recently and HER 2 which provides a specific treatment. All these are already known topics but which are willingly reread in a general examination. In our hospital we have conducted a study with good results in advanced gastric tumors (doi.org/10.1097/CAD.0000000000000877 to be cited in the bibliography) which perhaps adds something to the types of therapy proposed.”

 We revised the discussion regarding the options of neoadjuvant therapy (see above). The reference has been added to the bibliography of the revised manuscript.

  1. “We absolutely agree that to treat these patients it is necessary to organize multidisciplinary meetings to agree on the best diagnostic/therapeutic approach through brain storming; and during the established path, the discussion must be updated once the results are achieved. The paper must be reviewed in light of these lines to be accepted. The bibliography is good, the tables are explanatory.”

We addressed these considerations in the discussion of the revised manuscript.

  1. Comments on the Quality of English Language: “there are few errors to review”

 Several errors have been corrected. Moreover, the revised manuscript has been edited by the MDPI English language editing service.

Reviewer 2 Report

Comments and Suggestions for Authors Gastric cancer is still a very tough task for the oncological medical community. Increase incidence and poor prognosis even after different treatment options is the standard. Recent proposal of CRS , Hipec and Pipac for peritoneal gastric metastases could change the profile of this aggressive cancer stage. This study is intended to clarify the literature effort on this topic until now and propose an algorithm for treatment. Suggestions for authors • 170-200 The chapter Palliative laparoscopic Hipec should be posponed after Pipac chapter. Palliation is intended to treat terminal disease, at the end of gastric cancer history and therefore it should come at last. • 201 Neoadjuvant treatment: intraperitoneal plus sistemic chemotherapy (NIPS), for better comprehension • 252 Chapter about Pipac. 1- A table could be useful with different experiences, incidence of restaged to surgery patients and survival results. A way to justify and promote Pipac use 2- It could be of interest to stress the different use of Pipac through the time. At the begginig indicated only for untreatable patients (Marc Raymond ) until now that is suggested also with neoadjuvant and adjuvant indications. The necessity of randomized studies. • 352 could show, showed • 353 could have achieved, has been achieved "

Two errors :

286  Piapc instead of Pipac

371 Cylke instead of Cycle

Author Response

Reviewer 2:

“Gastric cancer is still a very tough task for the oncological medical community. Increase incidence and poor prognosis even after different treatment options is the standard. Recent proposal of CRS , Hipec and Pipac for peritoneal gastric metastases could change the profile of this aggressive cancer stage. This study is intended to clarify the literature effort on this topic until now and propose an algorithm for treatment.”

  1. 170-200 “The chapter Palliative laparoscopic Hipec should be posponed after Pipac chapter. Palliation is intended to treat terminal disease, at the end of gastric cancer history and therefore it should come at last.”

The order of chapters has been revised.

  1. 201 “Neoadjuvant treatment: intraperitoneal plus sistemic chemotherapy (NIPS), for better comprehension.”

The title of the chapter has been changed in the revised manuscript.

  1. 252 Chapter about Pipac. “A table could be useful with different experiences, incidence of restaged to surgery patients and survival results. A way to justify and promote Pipac use.”

An additional table has been added to the revised manuscript.

  1. 252 Chapter about Pipac. “It could be of interest to stress the different use of Pipac through the time. At the begginig indicated only for untreatable patients (Marc Raymond ) until now that is suggested also with neoadjuvant and adjuvant indications. The necessity of randomized studies.”

We absolutely agree with the reviewer. The considerations mentioned above have been added to the revised manuscript.

  1. 352 “could show, showed”

The manuscript has been revised.

  1. 353 “could have achieved, has been achieved”

The manuscript has been revised.

  1. Comments on the Quality of English Language: “Two errors: 286  Piapc instead of Pipac 371 Cylke instead of Cycle”

The two errors have been corrected.

Reviewer 3 Report

Comments and Suggestions for Authors

1. English writing is poor and very hard to read. Extensive editing is necessary before it can be considered for publication.

2. Please the abbreviation carefully. Do you want to use pmGC or pm GC? Also it is necessary to define this term when first usage.

3. Please give a short summary in each references you mentioned, instead of only providing the survival data. It is very helpful and crucial in every paragraph to address the survival benefit (or no benefit) of HIPEC or PIPAC.

4. Please check the recommeded regimen in Table 2. For example, Yonemura HIPEC regimen should use more latest one, like the reference No. 52, not early stage one.

Comments on the Quality of English Language

English writing is poor and very hard to read. Extensive editing is necessary before it can be considered for publication.

Author Response

Reviewer 3:

  1. “English writing is poor and very hard to read. Extensive editing is necessary before it can be considered for publication.”

The revised manuscript has been edited by the MDPI English language editing service.

  1. “Please the abbreviation carefully. Do you want to use pmGC or pm GC? Also it is necessary to define this term when first usage.”

The abbreviation pmGC is now used for the term ‘peritoneal metastatic gastric cancer’ throughout the revised manuscript and introduced as recommended in the introduction.

  1. “Please give a short summary in each references you mentioned, instead of only providing the survival data. It is very helpful and crucial in every paragraph to address the survival benefit (or no benefit) of HIPEC or PIPAC.”

We agree with the reviewer that the survival benefit plays a pivotal role for consideration of HIPEC or PIPAC therapy in patients with pmGC. Thus, we discussed this point in the revised manuscript and added information on survival benefit if applicable (RCTs).

  1. “Please check the recommeded regimen in Table 2. For example, Yonemura HIPEC regimen should use more latest one, like the reference No. 52, not early stage one.”

 We totally agree with the reviewer that table 2 does not contain a complete list of published HIPEC regimens for patients with pmGC and also contains former regimens. Nevertheless, the intent is to illustrate the variety of published HIPEC regimens. As mentioned in the manuscript in our opinion published data does not support clear recommendations. Regardless these considerations a more recent HIPEC regimen published by Yonemura et al. in 2020 has been added to table 2.

  1. Comments on the Quality of English Language: “English writing is poor and very hard to read. Extensive editing is necessary before it can be considered for publication.”

The revised manuscript has been edited by the MDPI English language editing service.

Round 2

Reviewer 1 Report

Comments and Suggestions for Authors

While maintaining the initial structure, the work has undergone important changes in the examination, especially when the two key topics such as HIPEC and PIPAC are addressed, which make it absolutely better than the first draft. The topics that I wanted to be examined and highlighted were taken up again in the discussion. I believe that the paper has reached the right maturity to be exposed to a wider audience

Reviewer 3 Report

Comments and Suggestions for Authors

I have no more question.